# ENDOG Impacts on Tumor Cell Proliferation and Tumor Prognosis in the Context of PI3K/PTEN Pathway Status

**DOI:** 10.3390/cancers13153803

**Published:** 2021-07-28

**Authors:** Gisel Barés, Aida Beà, Luís Hernández, Raul Navaridas, Isidre Felip, Cristina Megino, Natividad Blasco, Ferran Nadeu, Elías Campo, Marta Llovera, Xavier Dolcet, Daniel Sanchis

**Affiliations:** 1Departament de Ciències Mèdiques Bàsiques, Universitat de Lleida-IRBLleida, 25198 Lleida, Spain; gisel.bares@cmb.udl.cat (G.B.); aida.bea@udl.cat (A.B.); natividad_blasco@iislafe.es (N.B.); marta.llovera@udl.cat (M.L.); 2Lymphoid Neoplasm Program, Institut d’Investigacions Biomèdiques Agustí Pi i Sunyer (IDIBAPS) and CIBERONC, 08036 Barcelona, Spain; nadeu@clinic.cat (F.N.); ecampo@clinic.cat (E.C.); 3Departament de Ciències Mèdiques Bàsiques, Universitat de Lleida–IRBLleida and CIBERONC, 25198 Lleida, Spain; raul.navaridas@udl.cat (R.N.); ifelip@catsalut.cat (I.F.); cristina.megino@udl.cat (C.M.); xavi.dolcet@udl.cat (X.D.); 4Department of Oncology, Hospital Clinic of Barcelona, Universitat de Barcelona, 08036 Barcelona, Spain

**Keywords:** ENDOG, PTEN, AKT, endometrial carcinoma, glioblastoma, chronic lymphocytic leukemia

## Abstract

**Simple Summary:**

The PI3K/AKT pathway is involved in cell survival and proliferation. Molecular aberrations and/or hyperactivation of the PI3K-PTEN-AKT axis are frequent in distinct cancer types such as endometrial carcinoma, the most common type of cancer of the female genital tract, and chronic lymphocytic leukemia (CLL), a mature B-cell neoplasm depending on B-cell receptor (BCR) activity, which induces chronical activation of this pathway. The mitochondrial nuclease ENDOG was found to influence PI3K/AKT activity in somatic cells. Our aim was to assess the value of *ENDOG* gene silencing to block cancer cell proliferation and to evaluate the relevance of *ENDOG* as prognostic marker. In vivo, in vitro and in silico experiments show that *Endog/ENDOG* silencing blunts proliferation of tumor cells dependent on high *p*-AKT/low PTEN activity and that *ENDOG* has prognostic value in specific cancer types.

**Abstract:**

EndoG influences mitochondrial DNA replication and is involved in somatic cell proliferation. Here, we investigated the effect of *ENDOG/Endog* expression on proliferation in different tumor models. Noteworthy, *ENDOG* deficiency reduced proliferation of endometrial tumor cells expressing low PTEN/high *p*-AKT levels, and *Endog* deletion blunted the growth of PTEN-deficient 3D endometrial cultures. Furthermore, *ENDOG* silencing reduced proliferation of follicular thyroid carcinoma and glioblastoma cell lines with high *p*-AKT expression. High *ENDOG* expression was associated with a short time to treatment in a cohort of patients with chronic lymphocytic leukemia (CLL), a B-cell lymphoid neoplasm with activation of PI3K/AKT. This clinical impact was observed in the less aggressive CLL subtype with mutated IGHV in which high *ENDOG* and low *PTEN* levels were associated with worse outcome. In summary, our results show that reducing *ENDOG* expression hinders growth of some tumors characterized by low PTEN activity and high *p*-AKT expression and that *ENDOG* has prognostic value for some cancer types.

## 1. Introduction

Cancer is the second leading cause of death globally, accounting for more than 9 million deaths each year, and its burden continues to growth (WHO https://www.who.int/health-topics/cancer#tab=tab_1, accessed on 18 May 2021). The genetic heterogeneity and extreme diversity of the molecular bases sustaining the development of specific cancer types [1,2] require the identification of novel molecular contributors that support growth in some cancer types in order to improve their treatment [3].

The mitochondrial endonuclease ENDOG, which is involved in mitochondrial DNA (mtDNA) replication [4,5] and caspase-independent cell death [6,7], is required for normal somatic cell proliferation [8]. Lack of ENDOG increases mitochondrial ROS production and slows down cell proliferation in an ROS-dependent manner associated with reduction in the AKT/PKB-GSK-3-Cyclin D axis [8]. This finding adds further evidence on the important role of mitochondrial biology and mitochondria-derived molecules in the control of cytoplasmic transduction signaling that regulates many biological processes [9,10], including cell proliferation [11]. Moreover, AKT activation depends on the activity of PI3K, and it is antagonized by the tumor suppressor Phosphatase and Tensin Homologue (PTEN). In the context of cancer, it is well known that PI3K-PKB/AKT hyperactivation and alterations in PTEN [12] play a decisive role in the development and progression of many cancer types [13,14]. PTEN is one of the most frequently altered tumor suppressor genes in human cancer [15] by different mechanisms including mutation with loss of heterozygosity (LOH), promoter methylation or other processes that lead to its decreased expression [16]. Moreover, molecular aberrations in the PI3K-PTEN-AKT axis are very frequent (85–90%) in solid tumors, such endometrial carcinoma, the most common tumor type of the female genital tract [17]. In hematological cancer, this axis is fundamental in the pathogenesis of chronic lymphocytic leukemia (CLL), the most common leukemia in adults. This mature B-cell neoplasm fully depends on the B-cell receptor (BCR) activity for their proliferation and survival [18]. One of the key downstream elements of BCR is PI3K, which is chronically activated by the BCR stimulation even in the absence of genomic aberrations affecting the BCR signaling pathway [19,20].

In this context, we decided to investigate the potential role of ENDOG in tumor cell proliferation using different in vitro and in vivo cellular models and its potential clinical impact in CLL, which has an activated PI3K-PKB/AKT pathway in relation with their PTEN status.

## 2. Materials and Methods

### 2.1. Ethics Statement for Human Data

CLL expression data were obtained from previously published RNAseq data of patients from the ICGC-CLL Genome Project [21]. This cohort consisted of 266 CLL samples at diagnosis. All patients provided informed consent in accordance with requirements from local Institutional Review Board (Clinical Investigation Ethic Committee-CEIC, Hospital Clínic de Barcelona, Barcelona, Spain) (Project ICGC Genoma Leucèmia Crònica HBC;2009/5069) and following ICGC guidelines and Ethics and Policy Committee [22].

### 2.2. Animal Experimentation and Mouse Strains

The investigation with experimental animals was approved by the Experimental Animal Ethic Committee (CEEA) of the University of Lleida (codes CEEA06-01/10,07-01/10, 08–01/09 and 09–01/09), complying with the ARRIVE Guidelines and conforming to the Guide for the Care and Use of Laboratory Animals, 8th Edition, published in 2011 by the US National Institutes of Health. Mice were housed in Tecniplast GM500 cages (391 × 199 × 160 mm) never exceeding 5 adults/cage. All animals were housed at the Experimental Animal Housing Facility—University of Lleida, lights on from 7 a.m. to 7 p.m., temperature 18–22 °C and 30–70% humidity. Enriched environment included autoclaved cellulose material. Animals were fed 2914 diet (Irradiated Teklad Global 14% Protein Rodent Maintenance Diet, Harlan) and sterilized tap water, both ad libitum. The wellbeing of animals was monitored daily by visual inspection, and for SPF-housed mice, pathogen analysis was monitored from sentinel animals in periods of 8 weeks following the standards determined by the Federation of European Laboratory Animal Science Association (FELASA). Mice were sacrificed following the guidelines of our CEEA.

The *Endog* mouse colony is derived from founders given by Dr. Michael Lieber, University of Southern California, LA, CA, USA [23], and has a C57BL/6J background. Floxed homozygous *Pten* (C;129S4-Ptentm1Hwu/J; referred to here as *Pten^fl/fl^*) and Cre-ERT (B6. Cg-Tg (CAG-Cre/Esr1* 5Amc/J; referred to here as CAG-Cre) mice were obtained from the Jackson Laboratory (Bar Harbor, ME). *Endog^+/+^/Endog*^−*/*−^ CreERT^+/−^
*Pten^fl/fl^* mice were bred in a mixed background (C57BL6; 129S4) by crossing *Endog*^−*/*−^*, Pten^fl/fl^* and Cre-ERT^+/−^ mice. At 3 weeks after birth, animals were weaned and genotyped.

Immunodeficient SCID mice (age 12 weeks; weight 20–25 g) were subcutaneously injected with scrambled-transduced or *Endog*-specific shRNA-transduced IK cells (2 × 10^6^) suspended in 100 μL of PBS. Tumors were allowed to grow for 35 days. Tumors were measured every 2–4 days with calipers. Tumor size was calculated using the formula: tumor volume (mm^3^) = (D × d^2^)/2, where D = largest diameter and d = smallest diameter.

### 2.3. Cell Lines, Epithelial 3D Cultures and ENDOG/Endog Expression Silencing

The Ishikawa 3-H-12 (IK), HEC-1A and MFE-296 endometrial cancer cell lines were purchased from the American Type Culture Collection. Cal 62 and FTC 133 thyroid cancer cells were a gift from Dr. Santisteban (Instituto de Investigaciones Biomédicas Alberto Sols, Madrid, Spain). All cell lines were grown in Dulbecco’s modified Eagle medium (Sigma-Aldrich, St. Louis, MO, USA) supplemented with 10% fetal bovine serum (Invitrogen, Waltham, MA, USA), 1 mmol/L HEPES (Sigma-Aldrich), 1 mmol/L sodium pyruvate (Sigma-Aldrich, 2 mmol/L L-glutamine (Sigma-Aldrich) and 1% of penicillin/streptomycin (Sigma-Aldrichat 37 °C with saturating humidity and 5% CO_2_. U87-MG and U251-MG glioblastoma cells were obtained from CLS Cell Lines Service (Eppelheim, Germany). Glioblastoma cells were maintained in culture as indicated by the supplier at 37 °C and 5% CO_2_ atmosphere and were subcultured by trypsinization twice a week. These cell lines were previously characterized in our lab [24,25]. CaCO_2_ and HT-29 colorectal adenocarcinoma cells were provided by the Cell Culture Service from the University of Lleida. All cell lines were mycoplasma free (MycoAlertTM Mycoplasma Detection Kit, Cambrex, NJ, USA). All experiments were conducted with low-passage cells from recently resuscitated frozen stocks. Endometrium epithelial 3D cultures were established as described previously [26].

To reduce Endog expression in mouse cultured cells, we used a lentiviral vector system based in the plasmid pLVTHM (Dr. Trono’s laboratory, Switzerland) containing small hairpin RNA interference sequences targeting the mouse *Endog* mRNA sequence 5′-GGAACAACCTGGAGAAATA-3′, which was prepared as previously described [7]. For shRNA-mediated silencing of the human *ENDOG* gene, we used a commercially available pLKO.1 PURO plasmid (SHCLNG-NM_007931, Sigma-Aldrich). A lentiviral vector containing a scrambled sequence from the mouse sequence was used as control.

### 2.4. Cell Counting, Bromodeoxyuridine Incorporation and Cell Cycle Analysis

Equal numbers of cells were seeded from low-passage plates (lower than 20). Lentiviral transduced cells were incubated for 3 days, and after trypsinization, cells were seeded at equal densities. In every experiment, cells from 2 plates were counted a few hours after seeding to confirm equal initial cell numbers. Cultures were left to grow during 48–72 h. At the end of the period, plates were rinsed with PBS and trypsinized. After mild centrifugation, pellets were resuspended in PBS and counted in a Neubauer cell chamber under a phase contrast microscope. The exact number of independent experiments performed in duplicate is specified in the figure legends.

For drug treatment experiments, after 48h of lentiviral transduction, IK cells were trypsinized, and equal numbers of cells were seeded and left to attach for 24 h. The next day, drugs were added to culture plates: 10 µmol/L SAHA (Suberoylanilide hydroxamic acid/Vorinostat; Sigma-Aldrich; SML0061), 100 µmol/L Etoposide (Sigma-Aldrich; E1383) or vehicle DMSO (no drug, ND), and incubated for 24 h. Finally, cells were counted by Trypan blue exclusion as previously explained. The experiment was repeated three times.

The bromodeoxyuridine protocol was performed as described previously [26]. Briefly, 3D cultures were incubated with 3 ng/mL 5-bromo-2-deoxyuridine (BrdU) (Sigma-Aldrich; B5002) for 15 h and then fixed with 4% paraformaldehyde. After DNA denaturing with 2 mol/L HCl for 30 min and neutralization with 0.1 mol/L Na_2_B_4_O_7_ (pH 8.5) for 2 min, cells were blocked in PBS solution containing 5% horse serum, 5% fetal bovine serum, 0.2% glycine and 0.1% Triton X-100 for 1 h. Subsequently, cells were subjected to indirect immunofluorescence with a mouse anti-BrdU monoclonal antibody (diluted 1:100, DAKO, Carpenteria, CA, USA) and Alexa Fluor-conjugated anti-mouse secondary antibody. Nuclei were counterstained with 5 µg/mL Hoechst 33258, and cells were visualized under a confocal microscope. BrdU-positive nuclei were scored and divided by the total number of cells (visualized by Hoechst staining). The results are expressed as a percentage of BrdU-positive cells.

For cell cycle analysis, IK and U251 cells were seeded in 60 mm Petri dishes at 1.25–2.5 × 10^6^ cells/dish and incubated for 48 h. Afterward, cells were trypsinized and washed in ice-cold PBS, fixed in 70% ethanol and stored at –20 °C until analysis. Fixed cells were suspended in 500 μL propidium iodide (PI)/RNase staining buffer (Becton Dickinson, Franklin Lakes, NJ, USA), incubated for 30 min at 37 °C and analyzed in a BD FACSCanto II cytometer (Becton Dickinson) as in previous works [8,25]. Data analysis was performed using ModFit LT software (Verity software house, Topsham, ME, USA).

### 2.5. ROS Detection

ROS production was determined by using the fluorescent probe MitoSOX™ Red, a cationic derivative of DHE, which is rapidly and selectively targeted to the mitochondria where, as stated by the manufacturer, it is oxidized mainly by superoxide anions (O_2_^−^) (Thermo Fisher Scientific, Waltham, MA, USA; M36008). We have used previously MitoSOX™ Red to detect superoxide anions generated by mitochondria using flow cytometry [5,8]. Briefly, cells were washed twice in PBS, trypsinized and counted using a Neubauer cell chamber. Equal amounts of cells from each population were incubated in PBS containing 2.5 μmol/L MitoSOX™ Red for 10 min. Fluorescence of the cell population is proportional to the levels of intracellular ROS generated and was measured with a BD FACSCanto II cytometer (Becton Dickinson, Mountain View, CA, USA) using 488 nm laser excitation and detection with BP 585/42 filter.

### 2.6. Analysis of Protein Abundance

Protein extraction, SDS-PAGE in standard polyacrylamide gels and Western Blot were performed and analyzed as described elsewhere [7]. Antibodies were against human and rodent ENDOG (ab76122, Abcam, Cambridge, UK), PTEN, pS473-AKT (4060, Cell Signaling, Boston, MA, USA), AKT (sc-1618, Santa Cruz Biotechnology, Santa Cruz, CA, USA), pS21/9-GSK (9331, Cell Signaling), GSK3B (ab18893, Abcam), pY15-CDK1 (ab18, Abcam), CDK1 (ab32384, Abcam), Cyclin D (SC-20044, Santa Cruz Biotechnology), Cyclin B (ab181593, Abcam), Complex IV (Cox4; A21348, Thermo Fisher), Dihydrolipoamide DH (DLD; 133551, Abcam), CATENINB (610153, BD Biosciences, Franklin Lakes, NJ, USA), E-cadherin (610181, BD Biosciences) and Vimentin (ab45939, Abcam).

### 2.7. In Silico Datasets Exploration

Cancer Regulome explorer (http://www.cancerregulome.org/ (accessed on 26 January 2021) from The Cancer Genome Atlas Program, NIH) was used to investigate the potential association of *ENDOG* expression to any relevant characteristic such as histologic grade, gene expression and mutations in the available endometrial tumor dataset (ID: UCEC), whose latest available version is from June 2013. The results on the association between ENDOG expression and somatic mutations in uterine cancer are, in part, based upon data generated by the TCGA Research Network: https://www.cancer.gov/tcga (accessed on 26 January 2021) [27]. The distribution of ENDOG levels among normal endometrial tissues and different subtypes of endometrial tumors was obtained from UALCAN web resource (http://ualcan.path.uab.edu/) (accessed on 26 January 2021) [28].

### 2.8. Statistical Analysis

Statistical analysis of experimental data was performed using GraphPad Prism software (GraphPad Software, San Diego, CA, USA). Nonparametric tests were applied as follows: Mann–Whitney U test was used when comparing two independent data groups, and Kruskal–Wallis test was used when comparing more than two independent data groups, followed by the Dunn’s *post hoc* test for multiple comparisons and Dunnett’s test to compare against control. Analyses of cytometry data in Figure 1c and Figure 4c and data in Figure 1e,f,h were performed using a 2-WAY ANOVA test followed by Tukey’s post hoc test for multiple comparisons. Analysis of *ENDOG* expression data across histological subtypes in Figure 2e was performed through the UALCAN portal. The exact number of independent measurements and replicates are specified in each figure legend. All statistical tests were two-sided at a significance level of <0.05.

Overall survival and disease-free survival analyses on UCEC samples from the TCGA dataset from 2013 [27] were performed using R (version 3.6.3, R Foundation, Vienna, Austria). Optimal cut-off points for *ENDOG* expression groups were obtained using Maxstat algorithm, which optimized the log-rank statistics. Time to first treatment (TTT) of the CLL patients was calculated from sampling date to the time of first treatment. Optimal cut-off points for *ENDOG* and *PTEN* expression groups were obtained using the Maxstat algorithm. Cumulative curves of TTT and Gray’s test for the differences between the expression groups were performed using R (version 3.6.3). Dot plots for representation of *ENDOG* expression among the considered CLL subtypes in Appendix A and the corresponding unpaired Student *t*-tests to evaluate the differences in *ENDOG* mean expression were performed using GraphPad Prism software (GraphPad Software, San Diego, CA, USA).

Ranked lists of coding genes according to the degree of Pearson’s correlation to *ENDOG* expression were obtained for the different tumor subtypes in CLL. Only those statistically significant correlations (adjusted *p*-value < 0.05) after correction for multiple comparisons were considered. GSEA analyses were performed on those ranked lists to define the positively and negatively enriched molecular pathways (Hallmark and C2 curated gene signatures from MSigDB/GSEA website).

## 3. Results

### 3.1. ENDOG Gene Silencing Causes a Reduction in the Proliferation of Ishikawa Endometrial Adenocarcinoma Cell Line In Vitro and In Vivo in a Xenotransplant Mouse Model

In previous studies we had observed that ENDOG deficiency in somatic normal human and murine cells leads to reduced Akt phosphorylation (*p*-Akt) and reduced proliferation [8]. Therefore, we decided to assess the effects of silencing *ENDOG* expression on the proliferation of cancer cell lines and to determine its relationship with the levels of *p*-AKT expression. We silenced *ENDOG* expression in Ishikawa 3-H-12 (IK) cells, a tumor cell model of endometrial carcinoma, which is PTEN-deficient and has high *p*-Ser473-AKT expression [24,29]. We observed a significant decrease in tumor growth of ENDOG-deficient IK cells (Figure 1a). Cell death was always lower than 5% in the Trypan blue exclusion assays (Figure 1b) and flow cytometry experiments (Appendix A), regardless of the treatment. IK cells deficient for *ENDOG* expression accumulated in the S phase (Figure 1c). Likewise, reduced *ENDOG* expression induced a marked decrease in the phosphorylation of AKT and its downstream target GSK-3, as well as downregulation of β-catenin, cyclin D and cyclin B (Figure 1d; Appendix A), and it increased ROS production (Figure 1e). ROS scavenging with N-acetyl-L-cysteine (NAC) reduced ROS concentration in ENDOG-deficient IK cells (Figure 1e), but contrary to what we observed in somatic cells [8], it did not prevent the effects on proliferation (Figure 1f). Interestingly, when ENDOG-deficient IK cells were subcutaneously injected in the hind limbs of immune-deficient *Prkdc^scid^* mice, tumor growth was significantly reduced compared to control IK cells (Figure 1g). Finally, we assessed if ENDOG deficiency could affect the sensitivity to chemotherapeutic drugs such as SAHA (an HDAC inhibitor) and Etoposide (a DNA damaging agent). Both SAHA and Etoposide promoted cell death to a similar extent (approximately 70% decrease in cell survival) in Scr-transduced IK cells and cells expressing low levels of ENDOG (Figure 1h).

### 3.2. ENDOG Expression in Endometrial Cancer and Its Relationship to PTEN Status

Then, we analyzed the expression of ENDOG and several members of the PI3K/AKT pathway, both in several endometrial carcinoma cell lines and primary tumor databases, to investigate a possible association among them. *ENDOG* expression was directly correlated with AKT phosphorylation and inversely correlated with *PTEN* expression in a set of three endometrial adenocarcinoma cell lines. *ENDOG* expression was more abundant in IK cells, which have characteristics of endometrial carcinoma type I according to Bokhman’s pathogenetic descriptors [17]. On the contrary, HEC-1A and MFE-296 cells had low *ENDOG* expression (Figure 2a; Appendix A). HEC-1A and MFE-296 cells have high *PTEN* expression, which bears mutations in MFE-296 cells [29], and low AKT phosphorylation associated with low levels of β-catenin, E-cadherin and high vimentin expression; these are hallmarks of epithelium-to-mesenchyme transition (EMT) (Figure 2a), concordant with the more aggressive behavior of endometrial carcinoma type II, suggesting that low ENDOG is associated with a more aggressive phenotype. 

To further understand the relationship between ENDOG and PTEN alterations we performed an in silico search showing that *ENDOG* expression was lower in more aggressive uterine malignancies (Figure 2b; https://www.cbioportal.org/, accessed on 1 June 2021), and that *ENDOG* expression correlated exclusively with *PTEN* gene somatic mutations in endometrial cancer (TCGA Dataset: UCEC 28 June 2013 http://explorer.cancerregulome.org/, accessed on 1 June 2021) [27] (Figure 2c,d). Noticeably, *PTEN* mutations have previously been associated with the less aggressive histological subtype (endometrioid) of endometrial cancer [30]. Accordingly, *ENDOG* expression was found to be associated with a more favorable prognosis in endometrial cancer in TCGA data only for overall survival (*p* = 0.0062) (Appendix A), but not for disease-free survival (Appendix A). Similarly to *PTEN* mutations, high *ENDOG* expression levels were also found associated with endometrioid types, as 84.5% of patients with high expression had an endometrioid tumor (*p* < 0.00001; Fisher exact test), although no significant differences in overall survival could be found regarding *ENDOG* expression levels in any endometrial cancer subtype (Appendix A). These results supported that the clinical impact of *ENDOG* expression in endometrial cancer is a product of its association with the less aggressive subtypes. Taking into account the existing correlation between *ENDOG* expression and *PTEN* mutations, these results are also suggestive of a possible cooperation between both genes in the oncogenic transformation on some subsets of endometrial neoplasms. In accordance with this hypothesis, *ENDOG* expression showed significantly higher expression levels in endometrioid than in normal endometrial tissues, as well as compared to the more aggressive serous subtype (Figure 2e).

### 3.3. ENDOG Deficiency Reduces Proliferation of Polarized Endometrial Epithelial 3D Cell Cultures from Pten^−/−^ Mice

Loss of function-inducing genetic and epigenetic alterations in the tumor suppressor gene *PTEN* are common in many cancer types [13]. Because *ENDOG* silencing reduced proliferation of IK cells, a PTEN-deficient tumor cell line, we decided to further explore this event. We had previously developed a mouse model (CAG-Cre-ER^T+/−^*Pten*^fl/fl^)—in which tamoxifen (TAM) injection triggers *Pten* deletion mostly in epithelial cells, consistently inducing endometrial adenocarcinoma [31]—and a three-dimensional (3D) polarized endometrial epithelial cell culture model [26]. We induced lentiviral-driven shRNA-dependent *Endog*-specific silencing in 3D endometrial epithelial cell cultures from *Pten* transgenic mice in the presence or absence of the drug. TAM addition induced *Pten* deletion and enhanced gland growth in scrambled-transduced 3D cultures, but *Endog*-specific gene silencing strongly blocked TAM-induced gland growth (Figure 3a). Then, we assessed growth of 3D cultures obtained from *Endog*^+/+^/*Endog*^−/−^CAG-Cre-ER^T+/−^*Pten*^fl/fl^ female mice treated or not with TAM. In cultures expressing ENDOG, *Pten* deletion induced gland overgrowth, while lack of ENDOG blunted this effect (Figure 3b), accompanied by reduced abundance of *p*-Akt and cyclin D expression (Figure 3c; Appendix A), as well as reduced nuclear BrdU incorporation (Figure 3d). Together, these results show that ENDOG is required for endometrial epithelial cell overgrowth induced by *Pten* deficiency in vitro.

### 3.4. ENDOG Silencing Hinders Proliferation of Human Tumor Cell Lines with High Phosphorylation Levels of AKT and Low PTEN Expression

We next extended the analysis of ENDOG in cell proliferation to other relevant tumor types. First, we measured expressions of ENDOG, *p*-AKT, AKT and PTEN in FTC-133 follicular thyroid carcinoma cells, CAL-62 thyroid anaplastic carcinoma, HT-29 colorectal adenocarcinoma cells and U87 and U251 glioblastoma multiforme (GBM)-derived cancer cell lines. FTC-133, U87 and U251 cells showed the lowest levels of PTEN concomitant with the highest levels of *p*-AKT abundance (Figure 4a). Lentiviral-driven *ENDOG*-specific gene silencing reduced the proliferation of FTC-133 follicular thyroid carcinoma cells, but it had no effect on CAL-62 thyroid anaplastic carcinoma or HT-29 colorectal adenocarcinoma (Figure 4b). For the two GBM cell lines studied, *ENDOG* gene silencing caused a reduction in the proliferation of the U251 glioblastoma cell line, which showed cell cycle arrest in the S phase (Figure 4b,c). Therefore, in the cell lines tested, the effects of *ENDOG* silencing on cell proliferation occurred in tumor cells with the highest *p*-AKT and low PTEN, except for the U87 GBM cell line, pointing to the existence of additional factors determining the impact of ENDOG on proliferation. Altogether, these results further support the role of ENDOG in cell proliferation of tumors bearing low PTEN activity and high *p*-AKT abundance.

### 3.5. ENDOG Expression Is a Marker of Aggressiveness in a Subtype of Chronic Lymphocytic Leukemia (CLL) Patients, and Its Prognostic Value Is Dependent on Low PTEN Levels in These Patients

CLL is a frequent mature B-cell lymphoid neoplasm, where PI3K kinase is constitutively activated through BCR signaling [18], and may be an interesting tumor model to explore the possible clinical impact of *ENDOG* expression, either alone or in relation to *PTEN* status. We first analyzed ENDOG expression in our previously published cohort of CLL patients that has been well characterized by RNAseq and genomic studies in the frame of the International Cancer Genome Consortium (ICGC) project [22]. On this data set we defined an optimal cutoff for the clinical impact of *ENDOG* expression levels as detailed in the methods section. High *ENDOG* expression levels were significantly associated with short time from sampling to the need of treatment (time to first treatment, TTT), with a median of 1.97 years versus 13.61 years for patients with high and low *ENDOG* expression, respectively; *p* = 0.0006 (Appendix A). Two molecular CLL subtypes are distinguished based on the mutational status of the immunoglobulin heavy variable chain (IGHV), with cases carrying unmutated IGHV (U-CLL) having more aggressive behavior that CLL with mutated IGHV (M-CLL) [32,33]. Contrary to endometrial carcinoma, *ENDOG* median expression was slightly lower in the indolent CLL subtype (M-CLL) compared to the aggressive one (U-CLL) (*p* = 0.014) (Appendix A). *ENDOG* expression levels were associated with the evolution of M-CLL with a median TTT of 7.23 years versus 13.61 years for patients with high and low *ENDOG* expression, respectively (*p* = 0.0033) (Figure 5a) (Appendix A). No differences were seen in TTT of U-CLL patients in relation to low or high *ENDOG* expression (Figure 5a). 

We also analyzed the prognostic impact of *PTEN* in CLL. First, genomic data from the ICGC CLL project confirmed that *PTEN* mutations are rare in CLL (8/551), and their functional effect is questionable since only one mutation was found in coding regions, and it was a missense change (Appendix A). Next, we considered the impact of *PTEN* expression in the whole cohort and the two CLL molecular subtypes. *PTEN* median expression was higher in U-CLL than in M-CLL cases (*p* < 0.0001) (Appendix A). CLL patients with high *PTEN* expression had significantly shorter TTT than those with low expression of *PTEN* (median 0.74 years vs. 4.3 years for (*p* < 0.0001), Appendix A). However, no significant differences in TTT were observed between cases with high and low *PTEN* levels when analyzed separately in U-CLL and M-CLL (Appendix A and Appendix A). Regarding the interaction of *ENDOG* with *PTEN* in CLL subgroups, high *ENDOG* expression was associated with a significantly shorter TTT only in M-CLL with concomitant low *PTEN* expression levels (*p* = 0.0163) (Figure 5b and Appendix A). Thus, these results identify a subset of patients (ENDOG high/PTEN low) with inferior outcome inside the favorable prognosis in the M-CLL subtype.

### 3.6. ENDOG Expression Is Correlated with Genes in Pathways Related to Aggressiveness of CLL Including Cell Proliferation

To further understand the role of ENDOG in CLL pathogenesis, we performed Gene set enrichment analysis (GSEA) on genes ranked by their correlation with *ENDOG* expression in U-CLL and M-CLL. Interestingly, this analysis revealed that signatures previously related to CLL adverse prognosis [34] were significantly enriched in M-CLL but not in U-CLL (Appendix A). Regarding pathways related to cell proliferation, several were found enriched and involved genes either with positive or negative correlation to *ENDOG* in both CLL subtypes (Appendix A and Appendix A). Nevertheless, some particular cell-cycle related pathways were found exclusively enriched in M-CLL (Appendix A and Appendix A). Among the pathways positively correlated with *ENDOG* expression, *MYC* oncogene targets together with genes related to cell proliferation were observed. In this regard, the top *ENDOG*-correlated gene was *TBL3*, which downregulation has been related to a slower cell cycle [35] (Appendix A). Similarly, the top negatively *ENDOG*-correlated gene was *TCERG1*, known to be required for the growth arrest activity of C/EBPα [36] (Appendix A).

## 4. Discussion

The results presented here show that reducing *ENDOG* expression in human tumor cell lines with low *PTEN* expression and high *p*-AKT abundance restrains cell proliferation. We also report that ENDOG deficiency counteracts the stimulation of cell proliferation induced by *Pten* gene deletion in mouse endometrial epithelial cells. Finally, in silico data suggest that *ENDOG* expression associates with *PTEN* status in endometrial cancer and that its prognostic value in some CLL subtypes is also dependent on *PTEN* expression levels.

Hyperactivation of the PI3K/AKT signaling pathway is a frequent hallmark of human cancer [14], and development of related targeting drugs has shown to have great potential for cancer treatment [37]. However, due to the essential role of this signaling pathway for correct somatic cell function and to the resistance of some cancer types to treatment with available drugs inhibiting AKT activation, the identification and characterization of novel regulatory networks impinging on AKT activity can help to design and determine more efficient treatments. AKT activity depends on different regulators, including PI3K through the production of the PIP3 mediator from PIP2 in the plasma membrane. On the other hand, PI3K-dependent activation of AKT is also antagonized by PTEN, which dephosphorylates PIP3 to PIP2. Of note, AKT phosphorylation targets are involved in cell survival, growth, proliferation and other cellular processes relevant in cancer. Concordantly, molecular aberrations in the PI3K-PTEN-AKT axis are very frequent in many cancer types [12,13,14,17]. In this context, we recently showed the relationship of the mitochondrial nuclease ENDOG with cell proliferation, involving downregulation of the AKT/PKB-GSK-3-Cyclin D axis [8]. This finding suggested a potential wide impact of *ENDOG* expression in cancer, by which *ENDOG* could represent a new clinical biomarker and therapeutic target, and fostered our interest in assessing the impact of *ENDOG* expression in tumor cell proliferation.

Indeed, we show that *ENDOG/Endog* gene silencing reduces tumor cell proliferation in association with decreased *p*-AKT abundance and downregulation of the downstream signaling pathway. In addition, *Endog* silencing or genetic deletion can block the tumor promoting effects of *Pten* gene deletion in endometrial epithelial cells, paralleling the effects previously observed in somatic cells. However, some important differences have been observed in the processes altered by ENDOG deficiency in tumor cells compared to those observed previously [8]. While somatic cells tend to accumulate in the G1 phase, here we report that IK and U251 cells accumulate in the S phase after *ENDOG* gene silencing. Of note, IK cells have lost the expression of the tumor suppressors PTEN and retinoblastoma RB1 [29,38] and lack of RB1 facilitates E2F-dependent progression from G1 to S phase of the cell cycle. In fact, alterations in this pathway are found in most human cancers [39]. In addition, U251, but not U87, has a mutated p53 gene [40], which is another relevant component of the G1/S checkpoint. The absence of a functional G1/S checkpoint could explain the differences observed in cell cycle in the analyzed tumor cells compared to somatic cells that accumulate in G1 after *ENDOG* expression silencing.

Although ENDOG is involved in caspase-independent cell death in several somatic cell types [6,7,41] and is also involved in the sensitivity of some cancer cell lines to chemotherapeutic agents [42,43], our results discarded that changes in cell death contributed to the differences in cell number due to *ENDOG* silencing in the models used in this article. 

Previous reports showed that ENDOG contributes to the sensitivity to chemotherapeutic drug addition in breast [42] and prostate [43] cancer cell lines, measured as lactate dehydrogenase release to the culture medium. Our results show that *ENDOG* gene silencing did not block Ishikawa cell death induced by SAHA and Etoposide. The lowest cell number at the end of the treatment was attained by the combination of *ENDOG* gene silencing and drug treatment in our experimental models. 

In addition, although the effects on proliferation due to the reduction in *ENDOG/Endog* gene expression in somatic cells was ROS-dependent [8], this was not observed in IK endometrial carcinoma cells in which pharmacological ROS neutralization does not restore proliferation. In fact, ROS have been shown previously to induce proliferation in tumor cells [44,45]. These differences suggest that ENDOG controls diverse downstream events that impact proliferation in a cell type-dependent manner. The identification of the mitochondrial events regulated by ENDOG that impact AKT phosphorylation in tumor cells deserves further investigation. 

Our results show the correlation between the effect of *ENDOG* silencing on proliferation and the abundance of PTEN and *p*-AKT. In particular, in endometrial cancer cells, *ENDOG* silencing hampers proliferation of IK cells, which lack PTEN and have high levels of *p*-AKT. HEC-1A and MFE-296, which have more aggressive characteristics than IK cells and low *ENDOG* expression, have high expression of PTEN and low *p*-AKT abundance, despite harboring several mutations in the *PTEN* gene [29]. The results obtained from the analysis of *ENDOG* expression in the endometrial cancer databases showed a direct association of *ENDOG* expression and *PTEN* gene mutations and less aggressive histological subtypes. Although *ENDOG* expression was also found inversely associated with overall survival, this impact was not found to be independent of the observed association with the more indolent endometrial cancer subtype. However, information on the AKT activation status in the samples included in endometrial cancer datasets would be necessary to fully evaluate the *ENDOG* potential as a prognostic factor.

*ENDOG* gene silencing/suppression blunted AKT phosphorylation and the induction of cell proliferation in epithelial endometrial 3D cultures induced by PTEN gene suppression. These results demonstrated the involvement of the PI3K/AKT pathway in the effects of ENDOG in tumor cell proliferation in the assessed model. Analysis of the contribution of *ENDOG* expression to cell proliferation in diverse tumor cell lines with different status of the PTEN/*p*-AKT pathway further demonstrated that *ENDOG* silencing targets *p*-AKT-dependent signaling to reduce cell proliferation. Regarding the thyroid cancer cell models, the FTC-133 cell line, which has a hemizygous deletion of the *PTEN* gene and high *p*-AKT abundance [46], is sensitive to *ENDOG* silencing, whereas the CAL-62 cell line, which expresses high *PTEN* levels, is not. Intestinal epithelial tumor cell lines CaCO_2_ and HT29 have been shown to express *PTEN* and to require upregulation of *PTEN* and downregulation of PI3K activity for full intestinal cell differentiation [47,48]. However, our results show that, in these cells, *PTEN* expression does not inversely correlate to *p*-AKT abundance, as would be expected, suggesting the presence of molecular alterations and/or alternative signaling controlling AKT phosphorylation in these intestinal cancer cell lines. Nevertheless, ENDOG deficiency had some effect on HT29 cell proliferation, which has higher *p*-AKT abundance than CaCO_2_, supporting the notion that ENDOG effects on cell proliferation mainly course through the control of AKT activation. Finally, we considered the study of *ENDOG* silencing in U87 and U251 glioblastoma cell lines characterized, among other traits, by the absence of *PTEN* expression [40,49] and in which *PTEN* overexpression has been shown to sensitize to cell death inducers [50]. The differences between both cell lines regarding the effect on proliferation of *ENDOG* gene silencing could be due to the different levels of *p*-AKT combined with other genetic alterations. Of note, previous reports showed similar sensitivity of MFE-296, IK, CAL-62 and FTC-133 cell lines to the multikinase inhibitors Sorafemib and Regorafenib [24]. On the contrary, our results demonstrate that *ENDOG* gene silencing only influences proliferation in the tumor cell lines with the highest *p*-AKT abundance, which correlates with low or absent *PTEN* expression. This fact suggests that multikinase treatment most probably hampers other signaling pathways impacting in cell survival, in addition to the PI3K-AKT axis.

In this context we decided to explore the clinical impact of ENDOG in CLL as a model of B-cell neoplasm in which the PI3K-AKT axis is constitutively activated and required for cell survival and proliferation [18]. We analyzed the ENDOG expression in a series of 266 CLL patients from the ICGC project [21] and observed that its high expression levels were related to a more aggressive behavior with shorter TTT. A further dissection of the *ENDOG* expression impact on TTT in the two molecular subtypes of CLL could identify that *ENDOG* prognostic value was apparent only in the less aggressive M-CLL subtype. Interestingly, in the present study we also have found a relationship of *ENDOG* expression with less aggressive uterine malignancies in which the AKT pathway is hyperactivated. As we had observed that *ENDOG* expression correlated with mutations of the *PTEN* gene in endometrial cancer, we considered the relationship of *ENDOG* expression and *PTEN* status in CLL. *PTEN* has been previously found to be altered in cancer by different mechanisms including mutation with loss of heterozygosity (LOH), promoter methylation or other processes that lead to its decreased expression [16]. Although loss-of-heterozygosity of *PTEN* locus has been described in a proportion of CLL, PTEN mutations do not seem to be a relevant driver in CLL. Accordingly, the ICGC project only detected few cases with *PTEN* mutations, but their functional significance is questionable since they occur in introns and 3′-UTR. Low *PTEN* levels have been associated with shorter TTT in CLL, and other adverse prognostic factors [51]. In our CLL patients, low *PTEN* levels were related to a significantly shorter TTT only in patients with M-CLL subset with high *ENDOG* expression. Therefore, we conclude that high *ENDOG* levels have clinical impact in a CLL subgroup generally associated to a more indolent course but that confer a more aggressive behavior in those patients with concomitant low *PTEN* expression.

Finally, the pathway enrichment analysis on the transcriptomic data from the same CLL cohort studied for *ENDOG* clinical impact showed a significant enrichment in signatures previously related to poor prognosis [34], and those were found only in M-CLL, concordantly with the observed clinical impact of ENDOG in these CLL patients. Moreover, significant enrichments on several proliferation-related pathways, specifically in M-CLL, were also found, involving both positively and negatively *ENDOG* correlated genes.

## 5. Conclusions

In summary, the results presented in this work demonstrate that *ENDOG* gene silencing can be useful to reduce cell proliferation in some tumors characterized by low PTEN activity and high *p*-AKT abundance. Moreover, ENDOG expression associates with PTEN/*p*-AKT status in endometrial cancer and also has prognostic value in a tumor cell model with AKT hyperactivation, such CLL, but this depends on PTEN expression levels. Taking into account the high prevalence of alterations in the PI3K/AKT signaling pathway and the PTEN tumor suppressor in many types of tumors, our results pinpoint a new relevant factor to potentially improve prognosis and therapeutic approaches across some human neoplasms.

## Figures and Tables

**Figure 1 cancers-13-03803-f001:**
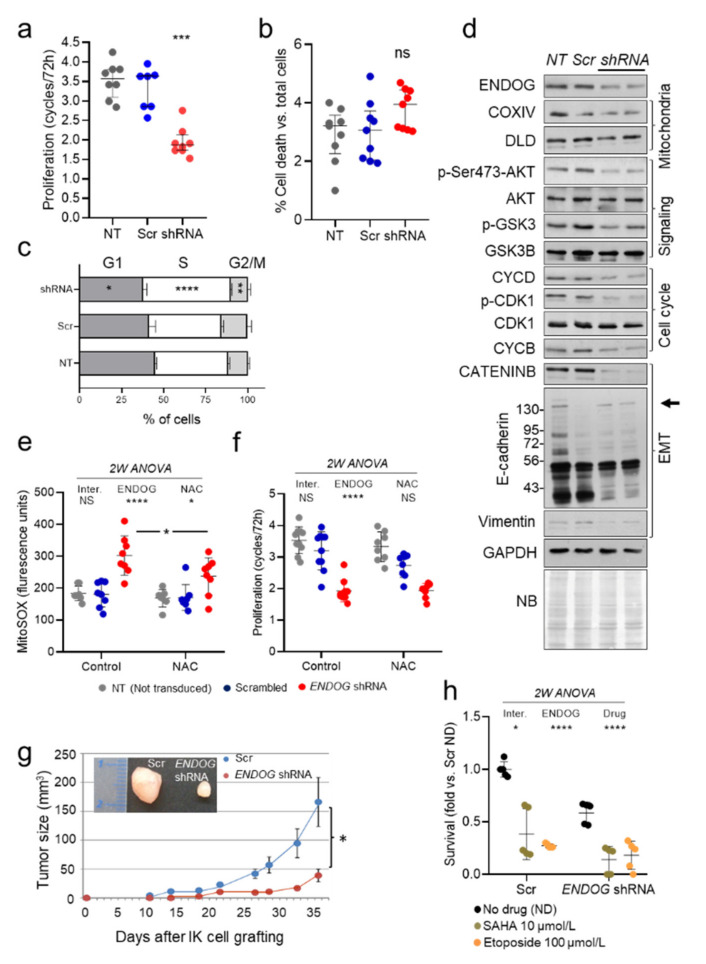
*ENDOG* silencing reduces proliferation in the Ishikawa endometrial carcinoma cell line in association with increased ROS production and reduced AKT-GSK-3 activation, hindering tumor growth. (**a**) IK cells were transduced with scrambled (Scr) or *ENDOG*-specific shRNA lentivirus or left to stand (Not transduced, NT). Equal numbers of cells were seeded in 2 plates/treatment and counted after 72 h. Data are expressed as the number of cell cycles completed in 72 h. Cells in replicate plates were counted at time zero to confirm equal initial cell numbers. All values from 4 independent experiments are plotted plus median ± interquartile range. Kruskal–Wallis test followed by Dunn’s test were performed. ***, *p* < 0.001 vs. Scr. (**b**) Percentage of dead cells (Trypan blue positive) counted in the experiments shown in (**a**). Medians ± interquartile range of *n* = 4 experiments are shown. Kruskal–Wallis test followed by Dunn’s test were performed. (**c**) Cells cultured in the same conditions as in (**a**) were detached at the end of the experiment, washed, fixed and stained with propidium iodide. The percentage of cells in each cell cycle phase was determined by flow cytometry. Data depicted on stacked bar graphs are mean ± SD. Statistical analysis was performed by 2-way ANOVA followed by Tukey’s test; *n* = 4 independent experiments in duplicates. *, *p* < 0.05; **, *p* < 0.001; ***, *p* < 0.001 vs. Scr. (**d**) Expression analysis of relevant genes was performed in total protein extracts of not transduced (NT), Scrambled-transduced (Scr) and *ENDOG* shRNA-transduced (shRNA) IK cell cultures. Mitochondria: COXIV subunit 4; DLD: Dihydrolipoamide dehydrogenase; Signaling: phosphorylated and total AKT and its substrate GSK3; Cell cycle regulators: CYCD: Cyclin D1; CDK1: Cyclin-dependent kinase 1; CYCB: Cyclin B. Epithelial–mesenchymal transition (EMT) markers: CATENINB, E-cadherin (molecular weight marks are added in kDa), Vimentin. Loading controls: GAPDH: Glyceraldehyde 3-phosphate dehydrogenase; NB: Naphthol blue staining of the membrane. Representative images of 3 independent experiments are shown. (**e**) ROS were detected using MitoSOX™ fluorescence quantified by flow cytometry in preparations of IK cells cultured under the same treatments as in (**a**) in the presence or absence of 0.2 mmol/L N-Acetyl-Cysteine (NAC; *n* = 4). All values from 4 independent experiments are plotted plus mean ± SD. 2-Way ANOVA followed by Tukey’s test; *, *p* < 0.05; ****, *p* < 0.0001. (**f**) Cell proliferation of cultures as in (**d**) was measured; *n* = 4 in duplicates. All values from 4 independent experiments are plotted plus mean ± SD. 2-Way ANOVA followed by Tukey’s test ****, *p* < 0.0001. (**g**) Eight 12-week-old SCID female mice were subcutaneously injected with 10^6^ cells/condition (scrambled, Scr; *ENDOG* shRNA) in their hind limbs (1 condition per limb). Tumor growth was measured with a digital vernier caliper. Animals were sacrificed before tumors reached a volume of 2.5 cm^3^. Tumor size: d^2^xD/2 (d and D: minor and major diameters). *n* = 5 mice. Data represent mean ± SD. Student t-test was performed to compare Scr and *ENDOG* shRNA tumors each day. (**h**) IK cells were transduced with scrambled (Scr) or *ENDOG*-specific shRNA lentiviruses and after 48 h, and equal numbers of cells were seeded. The next day, DMSO (no drug, ND), 10 µmol/L SAHA or 100 µmol/L Etoposide were added to cultures. Cell counting was performed 24 h after drug addition. Data are expressed as fold vs. Scr ND. Statistical analysis was performed by 2-way ANOVA followed by Tukey’s test; *n* = 3 independent experiments in duplicate. *, *p* < 0.05; ****, *p* < 0.0001.

**Figure 2 cancers-13-03803-f002:**
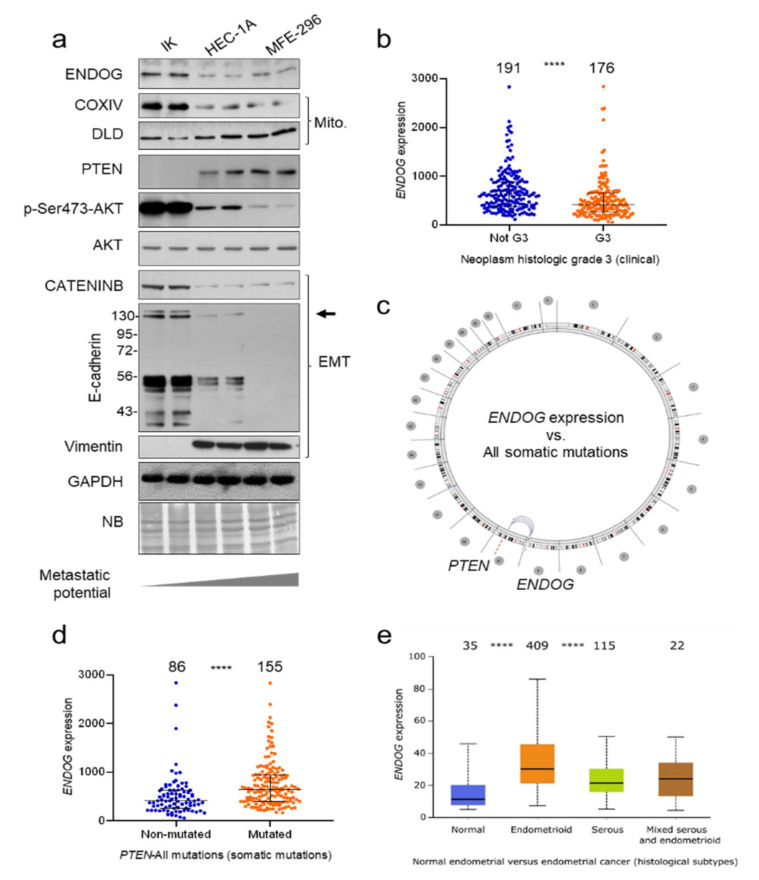
Study of *ENDOG* expression and its association with *PTEN* expression and mutations in cell lines and samples of endometrial cancer. (**a**) Expression analysis of relevant genes was performed in total protein extracts of IK, HEC-1A and MFE-296 endometrial adenocarcinoma cell lines. COXIV subunit 4; DLD: Dihydrolipoyl dehydrogenase. For E-cadherin, molecular weight marks are added in kDa, and the arrow on the right indicates the band of full-length protein. (**b**) Correlation of *ENDOG* expression with the histologic grade of endometrial cancer (carcinoma vs. adenocarcinoma) samples from cBioPortal datasets (https://www.cbioportal.org/). Sample size is indicated on the graph. Individual samples were plotted plus median ± interquartile range. (**c**) Whole genome search for association of *ENDOG* expression with somatic mutations on the Cancer Regulome Explorer in the UCEC dataset finds only a significant association with *PTEN* mutations (circular layout). (**d**) Tumor samples bearing mutations in *PTEN* have a higher level of *ENDOG* expression. Sample size is indicated on the graph. Individual samples were plotted plus median ± interquartile range. Mann–Whitney test: *p* < 0.0001. (**e**) Boxplots of *ENDOG* expression levels among normal endometrial tissues and the different histological subtypes of endometrial cancer (UCEC; TCGA project from UALCAN portal). *ENDOG* expression is significantly higher in the endometrioid than in the serous subtypes (*p* < 0.00001) or even normal endometrial tissues (*p* < 0.00001). ****, *p* < 0.0001.

**Figure 3 cancers-13-03803-f003:**
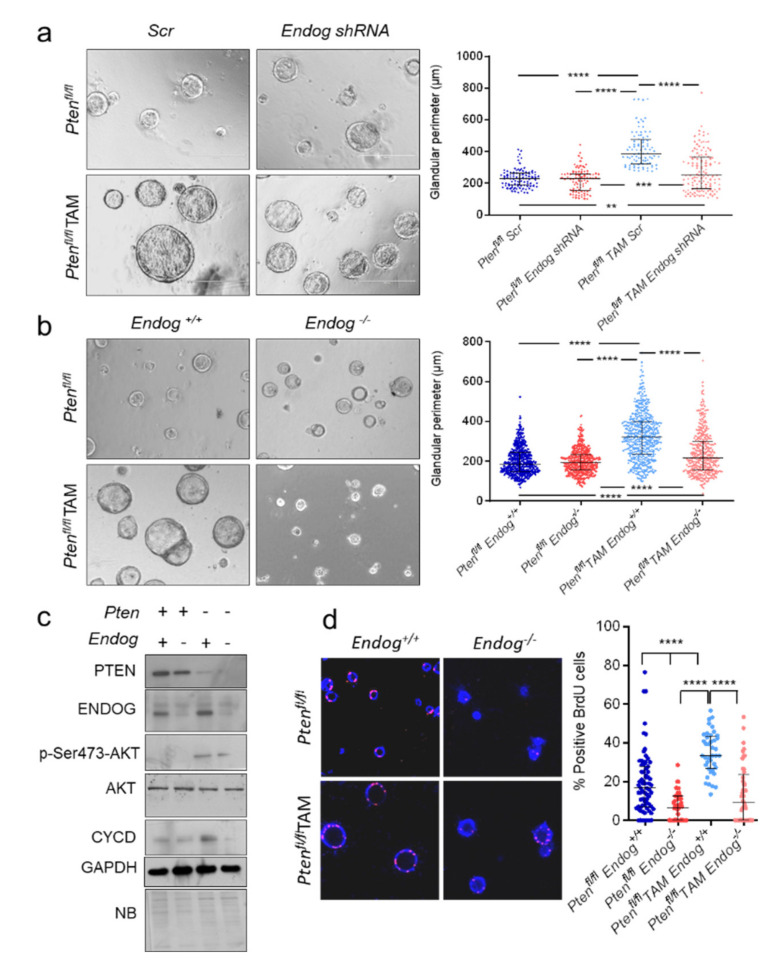
ENDOG deficiency prevents tumor growth induced by *PTEN* deletion in polarized endometrial epithelial 3D cell cultures. (**a**) Representative phase contrast microscopy images of *Pten*-expressing *Pten^fl/fl^* or *Pten*-deficient *Pten^fl/fl^* + tamoxifen (TAM) endometrial epithelial 3D cultures transduced with scrambled (Scr) or *Endog* shRNA lentiviral particles. Scale bar: 200 µm. Dot plot graph shows all individual measurements of the glandular perimeter obtained in an experiment performed in triplicate plates. Bars show median ± interquartile range. Kruskal–Wallis test followed by Dunn’s test was performed. ****, *p* < 0.0001. (**b**) Representative phase contrast microscopy images of endometrial epithelial 3D cultures obtained from *Pten^fl/fl^*/*Endog^+/+^* or *Pten^fl/fl^*/*Endog^−/−^* ± tamoxifen (TAM). Images are obtained at the same magnification as in (**a**). Dot plot graph shows all individual measurements of the glandular perimeter obtained in three independent experiments performed in duplicate plates. Bars show median ± interquartile range. Kruskal–Wallis test followed by Dunn’s test was performed. ****, *p* < 0.0001; ***, *p* < 0.001; **, *p* < 0.01. (**c**) Expression analysis of relevant genes was performed in total protein extracts of the cultures described in (**b**). NB: Naphthol blue staining of the membrane. (**d**) Representative fluorescence microscopy images of BrdU incorporation in endometrial epithelial 3D cultures obtained from *Pten^fl/fl^*/*Endog^+/+^* or *Pten^fl/fl^*/*Endog^−/−^* ± tamoxifen (TAM). Pink: BrdU, blue: Hoechst nuclear staining. Dot plot graph shows the percentage of BrdU-positive cells obtained in three independent experiments performed in duplicate plates. Bars show median ± interquartile range. Kruskal–Wallis test followed by Dunn’s test was performed. ****, *p* < 0.0001.

**Figure 4 cancers-13-03803-f004:**
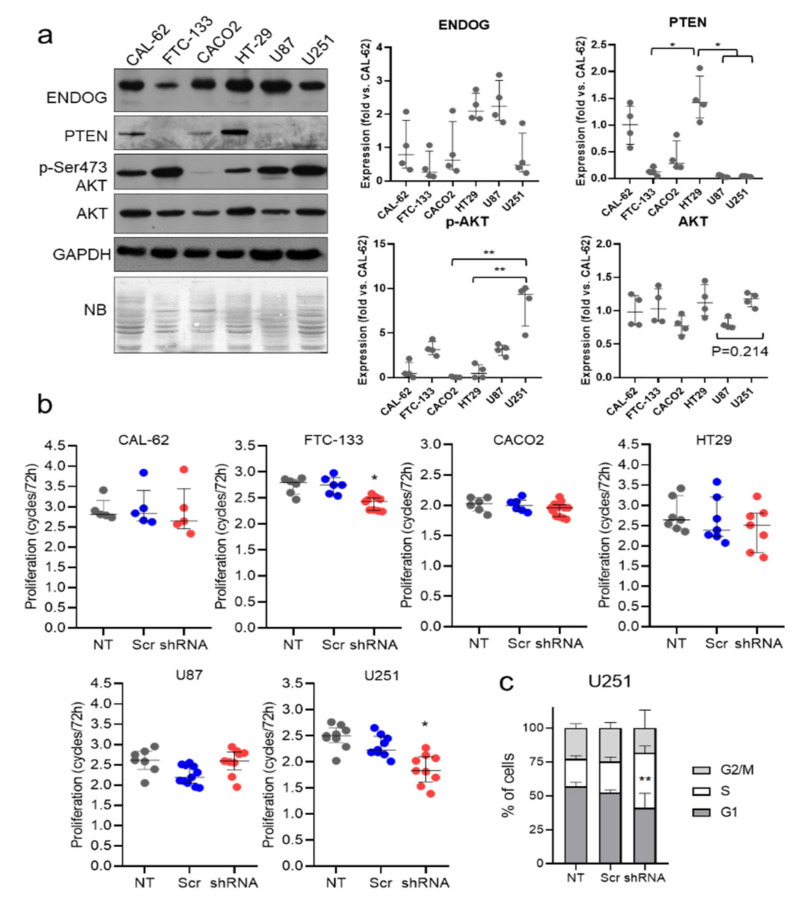
The efficiency of *ENDOG* expression silencing for reducing proliferation associates with the highest *p*-AKT and low PTEN expression in the thyroid, colorectal and glioblastoma tumor cell lines assessed. (**a**) Expression analysis of relevant proteins was performed in total extracts of CAL-62 thyroid anaplastic carcinoma, FTC-133 follicular thyroid carcinoma, CACO2 colorectal adenocarcinoma, HT29 colorectal adenocarcinoma and U87 and U251 glioblastoma cells. Representative blots of four independent sets of plates are shown. NB: Naphthol blue staining of the membrane. Graphs show densitometric analysis of Western blot images (relative individual values, median ± interquartile range). *, *p* < 0.05; ** *p* < 0.01 (**b**) The cancer cell lines in (**a**) were transduced with scrambled (Scr) or *ENDOG*-specific shRNA lentivirus or left to stand (not transduced, NT). Equal numbers of cells were seeded in 2 plates/treatment and counted after 72 h. Data are expressed as the number of cell cycles completed in 72 h. Cells in replicate plates were counted at time zero to confirm equal initial cell number. All values from 2 to 4 independent experiments performed in duplicate are plotted. Kruskal–Wallis test followed by Dunn’s test were performed. *, *p* < 0.05 vs. Scr. (**c**) U251 glioma cells cultured under the same conditions as in (**b**) were detached at the end of the experiment, washed, fixed and stained with propidium iodide. The percentage of cells in each cell cycle phase was determined by flow cytometry. Data depicted on stacked bar graphs are mean ± SD. Statistical analysis was performed by 2-way ANOVA followed by Tukey’s test; *n* = 4 independent experiments in duplicate. **, *p* < 0.01 vs. Scr.

**Figure 5 cancers-13-03803-f005:**
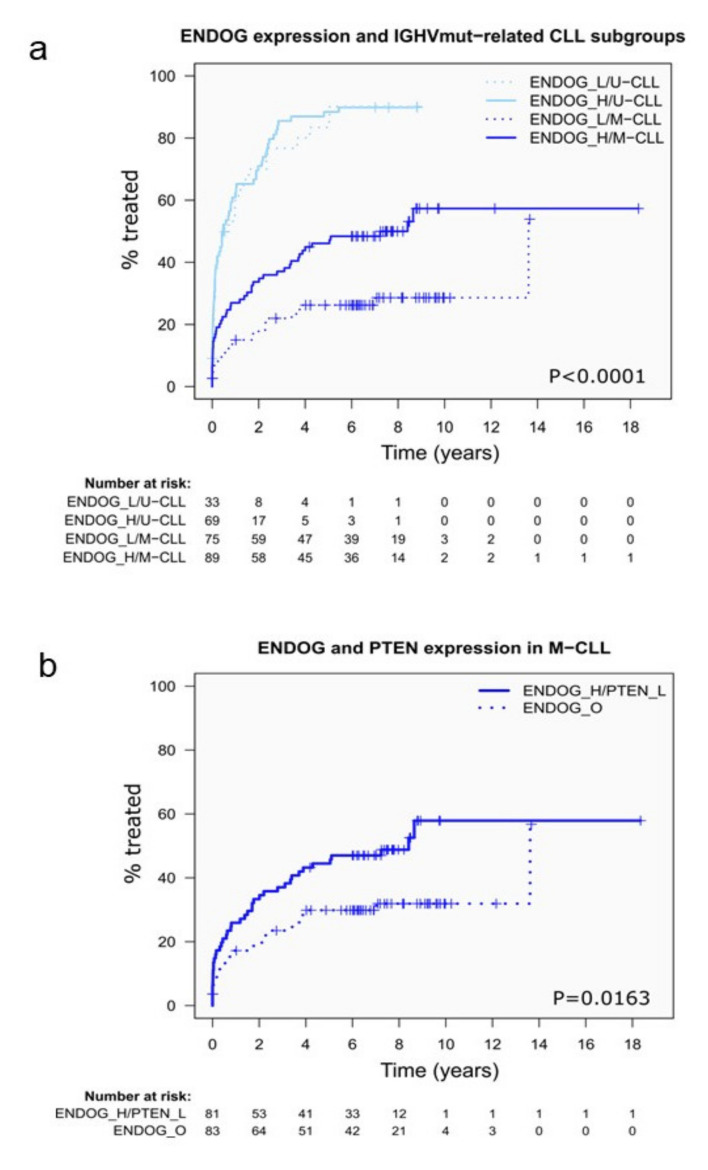
Time to first treatment (TTT) cumulative curves according to *ENDOG* expression in subgroups of CLL patients and their dependency on PTEN expression levels. (**a**) CLL patients with high (H) *ENDOG* expression levels showed a significantly shorter TTT compared to those with low (L) ENDOG expression levels, but only in the M-CLL subgroups between the categories defined by IGHV mutational status. (**b**) ENDOG expression association with a significantly shorter TTT in the M-CLL subgroup was only detected in cases with concomitant low PTEN expression levels compared to the other combinations of ENDOG and PTEN expression categories (O).

## Data Availability

Publicly available datasets were analyzed in this study. This data can be found here: Cancer Regulome explorer (http://www.cancerregulome.org/; TCGA Research Network: https://www.cancer.gov/tcga; UALCAN web resource (http://ualcan.path.uab.edu/. The data presented in this study about the ICGC-CLL Genome Project are available on request from Prof. Elías Campo at ECAMPO@clinic.cat.

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
