# Peer review of "ENDOG Impacts on Tumor Cell Proliferation and Tumor Prognosis in the Context of PI3K/PTEN Pathway Status"

_cancers, 2021, doi:10.3390/cancers13153803_

Round 1

Reviewer 1 Report

Few comments to further improve the manuscript:

1. > As per the Fig 1. legend - Fig 1b. supposed to be cell death data, based on trypan blue staining - this data is missing from the Fig 1.

> Fig 1 legend needs correction - Fig 1 (g) Animal experiment ; Fig 1(h) SAHA and Etoposide study

> Fig 1 ; Labeling needs correction; b->c ; c ->d; d ->e; e->f; f ->g; g ->h

> Fig 1(c) legend - change "Plates cultured in the same condition" to "Cells cultured in the same condition"

2. p-Ser(473)AKT (Methods, line 201; Fig. 1d; Suppl Fig 1b legend)

Authors need to change p-Ser(476)AKT to p-Ser(473)AKT in the manuscript; Indeed #4060, Cell Signaling antibody is p-Ser(473)AKT. 

3. Methods 2.2 Animal experimentation and mouse strains: Please include what "D" and "d" stands for in the tumor volume calculation (line 119).

Author Response

1. > As per the Fig 1. legend - Fig 1b. supposed to be cell death data, based on trypan blue staining - this data is missing from the Fig 1.

> Fig 1 legend needs correction - Fig 1 (g) Animal experiment; Fig 1(h) SAHA and Etoposide study

> Fig 1 ; Labeling needs correction; b->c ; c ->d; d ->e; e->f; f ->g; g ->h

> Fig 1(c) legend - change "Plates cultured in the same condition" to "Cells cultured in the same condition"

Authors: We regret the errors in Figure 1, which have been fixed in the current version. The Figure now presents cell death data in panel (b) and the figure legend has been updated.

2. p-Ser(473)AKT (Methods, line 201; Fig. 1d; Suppl Fig 1b legend)

Authors need to change p-Ser(476)AKT to p-Ser(473)AKT in the manuscript; Indeed #4060, Cell Signaling antibody is p-Ser(473)AKT. 

Authors: Thank you for the observation. We have corrected this problem.

3. Methods 2.2 Animal experimentation and mouse strains: Please include what "D" and "d" stands for in the tumor volume calculation (line 119).

Authors: We have added the requested information in the line.

Authors: We have had our MS revised and edited by a scientist not related to the article, with 5 years of postdoctoral experience in the USA and several years as corresponding author, in order to improve the quality of the text.

Reviewer 2 Report

Thank you for the revised version. However, I found that the authors did not improve the paper much. It is questionable for the reproducibility of the results if they refuse to provide another set of WB results. Also, the GAPDH added cannot represent the protein load of other target bands as they didn't revise all the protein bands simultaneously, i.e. GAPDH does not correspond with the other bands as they are conducted in different membranes. In addition, given the great number of protein targets conducted, there should be more loading controls but not only one for each WB round. Therefore, additional experiments needed.

Author Response

Reviewer 2.

Thank you for the revised version. However, I found that the authors did not improve the paper much. It is questionable for the reproducibility of the results if they refuse to provide another set of WB results. Also, the GAPDH added cannot represent the protein load of other target bands as they didn't revise all the protein bands simultaneously, i.e. GAPDH does not correspond with the other bands as they are conducted in different membranes. In addition, given the great number of protein targets conducted, there should be more loading controls but not only one for each WB round. Therefore, additional experiments needed.

Authors:

We regret that the reviewer doubts the integrity of some of our Western blot (WB) results. We did not repeat all the WB due to reasons explained in the initial rebuttal letter. The decision was based on the certainty that new WB of 3D culture experiments would not substantially improve the images already obtained due to inherent limitations of these experiments. However, this part of the manuscript is substantiated by the identification of the same trends in cell signaling in several experimental models (Fig.1c; 2a; 3c).

We share the reviewer’s point of view about the relevance of loading controls in WB. In the first version, we presented Naphthol blue (NB) staining of the same membranes used for immunoblotting. We think that NB staining is the best loading control. NB staining is performed on the same membranes used for blotting several antibodies. It gives an idea of the total protein loading, not only the expression of a single protein. We presented several considerations prompting us to prefer NB instead of analyzing the expression of “housekeeping” genes. Nevertheless, following the reviewer’s request, we performed GAPDH specific antibody staining of the membranes. GAPDH is broadly used as loading control in WB and also RT-PCR. We would like to point out that AKT is evenly expressed in all instances and could also be interpreted as loading control.

Finally, we have looked at loading controls used in many articles published in Cancers (Basel) and they have been performed in the style presented in the current version of our MS. Thus, in our opinion, our WB data and the conclusions derived from them are of similar quality than those published by the Journal.

Authors: We have had our MS revised and edited by a scientist not related to the article, with 5 years of postdoctoral experience in the USA and several years as corresponding author, in order to improve the quality of the text.

This manuscript is a resubmission of an earlier submission. The following is a list of the peer review reports and author responses from that submission.

Round 1

Reviewer 1 Report

In this manuscript Bares et al, investigated the effect of ENDOG expression on cell proliferation using various tumor models. Authors through in-vitro and in-vivo experiments demonstrated ENDOG silencing reduces Endometrial, colon, thyroid, and glioblastoma cell proliferation. Their results also showed a correlation between ENDOG inhibition on cell proliferation and the expression levels of PTEN and phospho-AKT levels. To further explore the prognostic value of ENDOG, authors analyzed CLL patient data set and showed ENDOG expression as a marker of aggressiveness in a subtype of CLL patients and its prognostic value is dependent on the PTEN status. Overall, this study provided a rationale to target ENDOG in various cancer patients with activated AKT signaling.

Major/Minor comments:

Fig 1D: Authors claim that NAC decreased the ROS levels induced by ENDOG suppression – but the data looks not convincing, there is no statistical significance.

Fig 1E: If ROS increase by ENDOG suppression is the reason for decrease in cancer cell proliferation – NAC should rescue this effect – As author mentioned this is not the case! Authors need to come up with a mechanism to explain why ENDOG suppression is decreasing the cell proliferation? Is there any change in apoptotic proteins with ENDOG suppression?

Fig 1C/Fig 2A: E-Cadherin blot is not clean – which band actually represents E-cadherin?

If ENDOG expression is directly correlate with p-AKT levels and cells respond well to ENDOG suppression – How do Endometrial cancer cells with constitutive AKT activation (For Ex: Myr-AKT expressing MFE-296 stable cells) respond to ENDOG suppression?

Fig 4A/4B: U87 and U251 Glioblastoma cells – Even though these 2 cell lines show no PTEN and activated p-AKT, only U251 cells responded to ENDOG shRNA – Why U87 cells with high ENDOG levels (compared to U251) showed no response to ENDOG shRNA?

I think the cell line panel is not enough to convince the point claimed in Figure 4 “ The efficiency of ENDOG expression silencing for reducing proliferation associates with high p-AKT and low PTEN expression in thyroid, colorectal and glioblastoma cell lines”. No significant difference was observed in colorectal cancer cells after ENDOG suppression.

Authors need to show if ENDOG suppression has any effect on Apoptotic pathway – in regards to various cancer models used in the manuscript. EndoG has been recognized as a key endonuclease in regulating apoptosis and necrosis (PMID: 17046751).

A prostate cancer study published in 2008 (PMID: 18565644) – showed that EndoG may be regulated by methylation of its gene promoter. The expression of EndoG correlated positively with sensitivity to chemotherapy and the silencing of EndoG by siRNA decreased the sensitivity of the cells to the chemotherapeutic agents in the EndoG-expressing prostate cancer cell lines. Is this also true with the cancer models that are used in the manuscript? ENDOG suppression has any effect on the chemotherapeuticS anticancer activity?

Reviewer 2 Report

The paper by Bares et al investigate on the role of ENDOG on tumor cell proliferation and tumor prognosis. I have the following concerns.

Major concerns

  1. Introduction part is not informative. While I could get the reasons for working on endometrial carcinoma and chronic lymphocytic leukemia. Why the study also investigate colorectal cancer, GBM and thyroid cancer cells?
  2. Fig.1C No loading controls. Please specify phosphorylation site of p-AKT.
  3. Fig.2A No loading controls. WB for ENDOG, COXIV, DLD are ugly. Please redo the experiments.
  4. Fig.3C No loading controls. WB for Pten, EndoG, p-Akt, CycD are ugly. Please redo the experiments, and also make the capitalization of protein targets uniform in different figures. Fig 3A,B,D. Graph resolution too low.
  5. Fig.4B No loading controls
  6. Fig 4A,B. Graph resolution too low.

Minor concerns.

  1. Line 112, should be thyroid cancer cells.
  2. The title should specify which type of tumor(s) that ENDOG impacts. It is a rather misleading title as ENDOG only impacts few cancer cell lines in the paper.

As the WB experiment results are not reliable, I suggest the paper to be rejected.

Reviewer 3 Report

The paper by Barés et al entitled “ENDOG impacts on tumor cell proliferation and tumor prognosis in the context of PI3K/PTEN pathway status ", presents the effect of ENDOG/Endog expression on proliferation in different tumors models and its relevance as a prognostic marker. The authors show that ENDOG deficiency negatively impacted the proliferation of endometrial tumor cells expressing low PTEN/high p-AKT levels as well as the growth of Pten-deficient 3D endometrial cultures. They also indicated that increased expression of ENDOG was associated with a short time to treatment in a cohort of patients with B-cell lymphoid neoplasm, which harbors PI3K/AKT activation.

They conclude that reducing ENDOG expression hinders the growth of tumors characterized by low PTEN activity and high p-AKT expression and that ENDOG 38 has prognostic value for some cancer types. This manuscript presents some interesting data that justify the conclusions and the manuscript is well written.

Comments and Suggestions for Authors

I have only one minor concern with this paper. Although the authors gave a very good rationale for the abrupt switch from studies with solid tumors to CLL, since the Authors conducted most of the experiments with endometrial cancer cell lines, why did they not look at the prognostic impact of ENDOG expression patients with endometrial cancer?